# Treatment-Related Adverse Events in Individuals with BRAF-Mutant Cutaneous Melanoma Treated with BRAF and MEK Inhibitors: A Systematic Review and Meta-Analysis

**DOI:** 10.3390/cancers17193152

**Published:** 2025-09-28

**Authors:** Silvia Belloni, Rosamaria Virgili, Rosario Caruso, Cristina Arrigoni, Arianna Magon, Gennaro Rocco, Maddalena De Maria

**Affiliations:** 1Department of Public Health, Experimental and Forensic Medicine, Section of Hygiene, University of Pavia, 27100 Pavia, Italy; cristina.arrigoni@unipv.it; 2Department of Biomedicine and Prevention, University of Rome Tor Vergata, 00133 Rome, Italy; r.virgili@idi.it; 3Department of Biomedical Science for Health, University of Milan, 20133 Milan, Italy; 4Health Professions Research and Development Unit, IRCCS Policlinico San Donato, 20097 San Donato Milanese, Italy; arianna.magon@grupposandonato.it; 5International Center for Nursing Research Montianum, Our Lady of Good Counsel Catholic University, 1000 Tirana, Albania; genna.rocco@gmail.com; 6Center of Excellence for Nursing Culture and Research, Order of Nursing Professions of Rome, 00136 Rome, Italy; 7Department of Life Health Sciences and Health Professions, Link Campus University, 00165 Rome, Italy; m.demaria@unilink.it

**Keywords:** cutaneous melanoma, clinical trials, vemurafenib, dabrafenib, trametinib

## Abstract

**Simple Summary:**

Cutaneous melanoma (CM) is a major public health concern, with rising morbidity and mortality worldwide. Advances in pharmacological science have led to the development of small-molecule kinase inhibitors (SMIs), particularly those targeting the mitogen-activated protein kinase (MAPK) pathway—B-Rapidly Accelerated Fibrosarcoma (BRAF) inhibitors and MAPK/extracellular signal-regulated kinase (MEK) inhibitors—which can directly inhibit melanoma cell growth in tumors with specific oncogenic mutations. While these agents are designed to selectively target cancer cells and limit toxicity, clinical trials have reported a range of treatment-related adverse events (TRAEs). Understanding the prevalence of these TRAEs is essential for optimizing symptom management and developing personalized risk-stratified care pathways. In this study, we systematically reviewed the literature and conducted a meta-analysis to quantify the incidence of TRAEs associated with the currently approved BRAF and MEK inhibitors in individuals with CM. The most frequently observed TRAEs included joint pain (arthralgia), skin rash, fever (pyrexia), fatigue, and the development of certain skin lesions, such as squamous cell carcinoma and keratoacanthoma.

**Abstract:**

**Objectives**: We conducted a systematic review of clinical trials and case reports analyzing the safety of the currently approved BRAF and MEK inhibitors in adults with cutaneous melanoma (CM), and a meta-analysis to estimate the pooled prevalence of treatment-related adverse events (TRAEs). **Methods:** We systematically searched six databases for studies published **since** 2009. The TRAE absolute frequencies reported in primary studies were aggregated using the Metaprop command in Stata 17, which calculates 95% confidence intervals (CIs) incorporating the Freeman–Tukey double arcsine transformation of proportions to stabilize variances within random-effect models. Methodological quality was assessed using the RoB 2 tool for randomized controlled trials (RCTs) and the ROBINS-I tool for non-randomized studies. **Results**: Twelve RCTs, **thirteen** prospective cohort studies (PCSs), and ten case reports were included. Meta-analysis was feasible for two regimens: vemurafenib 960 mg monotherapy and dabrafenib 150 mg twice daily plus trametinib 1–2 mg daily. The most common TRAEs during vemurafenib treatment were musculoskeletal and connective-tissue disorders (24%, 95% CI: 6–41%, *p* = 0.01), with arthralgia as the most prevalent (44%, 95% CI: 29–59%, *p* < 0.001), followed by rash (39%, 95% CI: 22–56%, *p* < 0.001). The most common TRAEs during dabrafenib plus trametinib were constitutional toxicities (classified in CTCAE as ‘General disorders and administration site conditions’; 25%, 95% CI: 14–37%, *p* < 0.001), with fatigue as the most prevalent (47%, 95% CI: 38–56%, *p* < 0.001), followed by pyrexia (40%, 95% CI: 26–54%, *p* < 0.001). Squamous cell carcinoma and keratoacanthoma were among the most frequent grade ≥ 3 cutaneous adverse events observed with vemurafenib therapy. **Conclusions**: Although additional large-scale studies are needed to corroborate these findings, each treatment has a distinct toxicity profile that should be considered when developing personalized risk-stratified treatment plans and in guiding healthcare resource allocation in melanoma care.

## 1. Introduction

Cutaneous melanoma (CM) is a significant public health concern globally, accounting for the majority of skin cancer-related deaths given its high metastatic potential [1,2]. CM is the most common variant of melanoma, originating from malignant transformations of pigment-producing melanocytes in the basal layer of the skin epidermis [3,4]. Over the past three decades, the global incidence of CM has steadily increased. In 2022, 331,700 diagnosed cases and 58,700 deaths occurred worldwide [1,5]. Population growth and ageing are expected to contribute to a significant rise in the incidence of melanoma diagnoses, with a projection of almost 510,000 (50% increase) new cases and 96,000 (68% increase) deaths by 2040 [6].

These trends required advances in pharmacological and pharmaceutical science and disease pathogenesis knowledge to determine the best treatment option [2]. As a result, the landscape of available therapeutic options has rapidly changed in recent years with the development of selective drugs [2]. Before 2010, radiotherapy and chemotherapy were the sole alternatives to surgery. Additional approaches such as interferon (IFN), high-dose interleukin-2 (IL-2), and biochemotherapy regimens were also employed, but they were associated with considerable toxicity and only transient benefits in most responders. Subsequently, small-molecule kinase inhibitors and immune checkpoint inhibitors were approved, transforming the therapeutic landscape [7]. Vemurafenib was the first BRAF inhibitor (BRAFi) approved by the Food and Drug Administration (FDA) for patients with advanced-stage melanoma in 2011 [8]. Two years later, additional BRAF inhibitors (dabrafenib and encorafenib) and MEK inhibitors (MEKis) (trametinib, cobimetinib, and binimetinib) were developed [2,9], showing higher benefits in terms of side effects and prognosis outcomes [10,11,12]. These advances increased survival rates among individuals with CM, resulting in growing healthcare demand [13,14].

Although the primary curative approach for early-stage localized CM is surgical excision, metastatic melanoma is rarely amenable to surgery due to the typically extensive metastatic spread, and systemic therapies remain the cornerstone of treatment [15,16]. CM presents a high rate of genetic mutations, with 70% of identified cases attributable to mutations in genes associated with primary signaling pathways [2,9,17]. These oncogenic mutations activate cellular signaling pathways, resulting in the uncontrolled proliferation of cancerous cells [18]. Mutations in the BRAF, NRAS, and NF1 genes are the most frequent and occur in approximately 50, 25, and 14% of melanoma cases, respectively [18]. The mitogen-activated protein kinase (MAPK)/extracellular signal-regulated kinase (ERK) signaling cascade plays a central role in melanoma pathogenesis, with activating mutations in BRAF and NRAS representing the most frequent molecular drivers [19]. Therefore, CM primarily benefits from inhibitors targeting BRAF and MAPK kinase (MEK) mutations, which specifically halt widespread proliferation [18].

Despite the unquestionable progress, treatment approaches currently remain limited in some cases, with the development of resistance mechanisms to therapies and an increased incidence of adverse side effects. Approximately 15–20% of CM cases exhibit innate resistance to this therapy, and 50% of patients frequently develop acquired resistance after approximately six months [20]. The adverse effects are largely attributable to off-target inhibition of the mitogen-activated protein kinase (MAPK) pathway in normal tissues. This signaling cascade is physiologically active in healthy cells, where it regulates proliferation, differentiation, and survival. Its systemic inhibition disrupts normal cellular processes and may give rise to toxicities affecting the gastrointestinal, musculoskeletal, cutaneous, metabolic, and cardiovascular systems [21,22].

Although the combination of BRAFis/MEKis reduces the overall incidence of toxicities compared with BRAFi monotherapy [23], it primarily extends time to resistance, prolonging both overall survival and the duration of treatment exposure. However, these side effects, although often manageable, may still compromise treatment continuity, requiring dose adjustments, temporary interruptions, or treatment changes, with potential impacts on clinical outcomes [24] and quality of life [25,26]. A systematic understanding of the aggregate prevalence of adverse events related to targeted therapies in patients with cutaneous melanoma is therefore essential for optimizing symptom management, guiding supportive interventions, and designing personalized risk-stratified care pathways. Previous meta-analyses have examined the safety of targeted therapies for melanoma [27,28,29]; however, these studies included individuals with both cutaneous and non-cutaneous melanoma, as well as a combination of different treatments, such as chemotherapy and immune checkpoint inhibitors. To address this gap, we aimed to synthesize clinical trials investigating the safety of the currently approved targeted therapies (BRAFis and MEKis) in adults with BRAF-mutant cutaneous melanoma and to quantitatively summarize the incidence of the most prevalent treatment-related adverse events (TRAEs). This meta-analysis therefore focuses on the most extensively studied targeted therapies—vemurafenib, dabrafenib, and trametinib—while other approved combinations, including vemurafenib plus cobimetinib and encorafenib plus binimetinib, could not be fully analyzed due to the limited number of eligible trials available at the time of our search.

## 2. Materials and Methods

This systematic review (SR) was registered on PROSPERO (CRD420250654016) on 26 March 2025 to ensure transparency in the review process, avoid duplications, and enable the comparison between the protocol and the Materials and Methods section of the manuscript [30]. Any deviation from the protocol was documented in the form of amendments in PROSPERO. This systematic review was conducted in accordance with the “Cochrane Handbook for Systematic Reviews of Interventions” [31]. Although the Cochrane handbook does not provide specific guidance for proportional meta-analyses, it describes how to meta-analyze various types of data, including incidences. The review reporting was conducted in line with the Preferred Reporting Items for Systematic Reviews and Meta-Analyses (PRISMA) statement (Appendix A) [32].

### 2.1. Eligibility Criteria

The Condition, Context, Population (CoCoPop) framework provided the basis for the main eligibility criteria: Co) TRAEs, Co) targeted therapies (BRAF inhibitors and MEK inhibitors), and Pop) adults aged ≥18 with CM. This framework is recommended for formulating research questions in proportional meta-analyses [33]. This framework reflects the study’s research question.

In addition to the elements of the CoCoPop framework, we applied the following inclusion criteria: (a) scientific methodological peer-reviewed papers, (b) any type of clinical trial reporting the incidence of TRAEs (as both primary or secondary outcome) associated with the approved target therapies (BRAF inhibitors and MEK inhibitors) for CM, (c) published between 2010 and the date of search, (d) containing the abstract and full text to be able to assess the methodological quality, and (e) original research articles published in any language. We also included case reports, which may provide pertinent insights for systematic review recommendations regarding new medications [34]. Both randomized and non-randomized studies were included to provide a comprehensive overview of AEs based on the study designs [3,35,36]. Letters to the editor, conference proceedings, poster presentations, and ongoing registered clinical trials were excluded from the selection because it was not possible to assess the quality of evidence. Retrospective studies were also excluded due to the intrinsic study design biases, such as recall bias, selection bias, and confounding factors, that can affect the accuracy and reliability of the results. Case reports were considered for qualitative synthesis only as they may highlight rare or unexpected toxicities not captured in larger trials. Conversely, targeted therapy regimens for which fewer than two eligible studies were available (e.g., encorafenib plus binimetinib; vemurafenib plus cobimetinib) were not meta-analyzed to avoid producing unreliable estimates.

### 2.2. Search Strategy and Data Sources

The Peer Review of Electronic Search Strategies (PReSS) guidelines were followed to improve the quality of the literature search [37]. In this regard, the recommended PRESS 2015 Guideline Evidence-Based Checklist was applied to examine the appropriateness of the search strategy. Firstly, the search strategy was created for the PubMed database using MeSH and free-text words; then, it was adapted for the other databases as appropriate. An author developed the search strategy, and it was peer-reviewed by another to achieve the optimal balance between sensitivity and specificity (Appendix A).

We searched the following databases for articles published between 2009 and the date of the search: PubMed, CINAHL, Web of Science, EMBASE, Cochrane Central Register of Controlled Trials (CENTRAL), and Scopus. The publication date filter, which limits the search to papers published from 2009, was configured considering the recent history of BRAF inhibitors and MEK inhibitors. The first phase I clinical trial of vemurafenib for BRAF mutant melanoma began in 2008 and was published in 2011 [38]. No language or any other restrictions were set in the database search. Additional methods, such as hand-searching and reference checking, were applied to maximize findings. The reference lists of the included full-text papers were manually examined, and forward citation analysis was conducted. No additional sources, such as Google Scholar, were used as it is primarily used to identify gray literature, which is unsuitable for our investigation [39].

### 2.3. Article Selection and Data Extraction

The CoCoPop framework and the eligibility criteria guided the article selection process. The article screening followed the PRISMA flow diagram [32,40]. Records were exported from the databases to the reference manager software (Zotero 7.0.16 for Windows), where duplicates were removed. After removing duplicates in Zotero, the records were imported into the Rayyan software (Rayyan Systems Inc., https://www.rayyan.ai/), and two authors independently screened the titles and abstracts of identified papers. After comparing the lists of eligible studies, the two authors performed the full-text evaluation. No disagreements were noted between the two authors during the screening and selection of the articles.

Two authors extracted data independently using a piloted electronic extraction form (Excel form) [31]. This approach enabled the inclusion of all relevant variables to facilitate the interpretation of the findings. Pertinent information was extracted from each eligible study, including first author/year, trial name, trial phase, study design, number of patients, treatment arms, molecular subtype, site of metastasis, AE assessment scale, and coding. Incidence data from each primary study were extracted in a separate table for meta-analyses. No substantial discrepancies emerged between the two authors, necessitating the involvement of a third reviewer. In cases where relevant information was missing for data analysis, the authors of the primary studies were planned to be contacted by email.

For each included trial, we recorded the starting dose and, when available, the frequency of dose-limiting toxicities (DLTs) and treatment discontinuations due to toxicity. Reported adverse events were generally cumulative across the duration of therapy, unless otherwise specified in the primary studies. In line with the Common Terminology Criteria for Adverse Events (CTCAE), the category ‘general disorders’—as reported in the primary studies and maintained in our tables—was defined in the text as constitutional toxicities. This category includes fatigue, asthenia, pyrexia, headache, chills, influenza-like illness, and peripheral edema.

### 2.4. Quality Appraisal

The Revised Cochrane Risk of Bias tool and the Risk of Bias in Non-Randomized Studies of Interventions (ROBINS-I) assessment tools were applied to evaluate studies’ risk of methodological bias in RCTs and non-randomized intervention studies, respectively [41,42]. This evaluation was conducted independently by two authors; the involvement of a third author was not necessary as the two authors agreed on the methodological quality of each study.

The revised RoB 2.0 encompasses five domains as follows: (1) bias arising from the randomization process, (2) bias due to deviations from intended interventions, (3) bias due to missing outcome data, (4) bias in the measurement of the outcome, and (5) bias in the selection of the reported result. Each domain comprises signaling questions to determine the risk of bias (“yes”, “probably yes”, “probably no”, “no”, “no information”, or “not applicable”). Responses to signaling questions serve as the foundation for domain-level evaluation: (a) Low risk of bias, (b) Some concerns, or (c) High risk of bias.

The ROBINS-I tool includes seven domains in total. Two domains refer to a pre-intervention assessment: (1) bias due to confounding and (2) bias in the selection of participants. One domain at the intervention assessment: (3) bias in the classification of interventions. Four domains at post-intervention assessment: (4) bias due to deviations from intended interventions, (5) bias due to missing data, (6) bias in the measurement of outcomes, and (7) bias in the selection of the reported result. Responses to signaling questions (“yes”, “probably yes”, “probably no”, “no”, “no information”, or “not applicable”) enable risk of bias judgment: (a) “Low” (b) “Moderate”, (c) “Serious”, (d) “Critical”, or (e) “No information”.

### 2.5. Data Synthesis

Data was analyzed using STATA 17 software (StataCorp. 2019. Stata Statistical Software: Release 17. College Station, TX, USA: StataCorp LLC). We quantitatively pooled the absolute frequencies reported in primary studies using the Metaprop command, which is the recommended approach for performing a proportion meta-analysis with binomial data [43]. We aggregated the data incidences if there were at least two studies for each outcome, and the study sample size was ≥25 individuals, to provide reliable results [44]. The *Metaprop* command estimates a 95% confidence interval (CI) using exact binomial and score test-based confidence intervals, incorporating the Freeman–Tukey double arcsine transformation of proportions [43]. This method utilizes the binomial distribution to forecast within-study variability, providing acceptable confidence intervals for each study and the aggregated prevalence [43]. Statistical significance was determined as a two-sided *p*-value of <0.05. Given the intrinsic heterogeneity of epidemiological trials, a random-effect model was used for the magnitude estimations [45].

After calculating the single-outcome prevalence for each study’s design group, categorized by treatment type, we quantitatively aggregated the pooled effect size of each outcome with 95% CI to estimate the magnitude of prevalence of the most common TRAEs. Given the high clinical heterogeneity between studies in terms of outcomes and the probability of real variations in the between-study proportion, a random-effect model of the inverse variance was adopted [46], applying the restricted maximum likelihood (REML) method [47]. The magnitude of inconsistency between studies was estimated by chi-squared (Q) and I^2^ statistics (I^2^) [48]. Statistical heterogeneity was defined as Q > df with *p* < 0.05 (“Q” is the chi-squared statistic, and “df” is its degrees of freedom) and I^2^ < 25% (low heterogeneity), between 25% and 50% as moderate, and >50% as high heterogeneity [48]. For each adverse event, pooled prevalence estimates with 95% CIs were calculated, and *p*-values indicate whether the pooled prevalence significantly differed from zero. Given the intrinsic heterogeneity in prevalence studies, we focused our interpretation on the pooled prevalence estimates and their CIs while reporting *p*-values for completeness. Between-study heterogeneity was quantified using the I^2^ statistic, which reflects the proportion of variability across studies due to heterogeneity rather than chance. As recommended, I^2^ values in meta-analyses of proportions should be interpreted with caution since high values are often driven by underlying clinical and methodological variability.

## 3. Results

A total of 1123 records were identified through searches in major bibliographic databases. Before initiating the screening process, seventy duplicates were removed, along with one retracted study. Following title and abstract screening of the remaining 1052 records, 979 were excluded because they did not meet the inclusion criteria. The remaining 78 articles were retrieved in full text for eligibility assessment. An additional three articles were identified through manual screening of the reference lists of the included studies. During the full-text screening phase, three articles were excluded [49,50,51] as they analyzed the same patient cohorts already included in three other eligible studies [51,52,53] without providing any additional relevant data. These articles were therefore considered duplicative analyses and not eligible for final inclusion. Overall, our systematic review included thirty-five articles: twelve RCTs [22,38,52,53,54,55,56,57,58,59,60,61], thirteen prospective cohort studies (PCSs) [62,63,64,65,66,67,68,69,70,71,72,73,74], and ten case reports [75,76,77,78,79,80,81,82,83,84]. A detailed description of the study selection process, including the reasons for article exclusion, is available in Figure 1.

### 3.1. Characteristics of the Studies

The included studies were published between 2011 and 2024. Eight studies were international multicenter trials [38,53,55,57,58,59,60,61]; five were conducted in the United States [22,66,69,70,74], two in Belgium [63,71], two in Spain [56], one in Italy [64], one in the United Kingdom [54], one in Australia [62], and two in East Asia [72,73]. Eight studies were phase III trials [38,52,53,55,57,58,60], eleven were phase II trials [22,54,56,61,63,67,68,69,71,72,74], and three were phase I/II or phase Ib studies primarily focused on evaluating novel treatment combinations or alternative dosing regimens [65,70,73]. All the studies enrolled adult patients (>18 years) with unresectable locally advanced or metastatic melanoma (predominantly stage IIIC–IV) harboring a confirmed BRAF V600 mutation, identified primarily through PCR-based assays, with some studies employing next-generation sequencing (NGS) or Sanger sequencing. The involved treatments were BRAF/MEK inhibitors used as monotherapy (vemurafenib and trametinib) or in combination (encorafenib plus binimetinib, dabrafenib plus trametinib, and vemurafenib plus cobimetinib). Adverse events were identified and graded in nearly all the studies using the CTCAE, primarily versions 3.0, 4.0, or 4.03. The main characteristics of the included studies are summarized in Table 1.

### 3.2. Pooled Prevalences of TRAEs

The meta-analysis was conducted based on the study design as the pooled effect size and interpretation may depend on methodological variations between randomized and non-randomized studies (i.e., different methods for measuring exposure and outcome, or adjusting for confounding domains). However, the meta-analysis approach excluded a subset of the included studies as the number of studies for each research design and combination therapy, and the sample sizes were, in some cases, inappropriate. The data from Dummer et al. on encorafenib monotherapy and its combination with binimetinib [55], as well as the data from Drèno et al. on cobimetinib and vemurafenib [53], were excluded from the meta-analyses of randomized controlled trials as they were the sole studies available for each treatment regimen. Regarding the PCSs, two articles focused on the efficacy and safety of vemurafenib in combination with cobimetinib [51,70]. However, these findings were part of the same clinical trial, and, to avoid overlap among the results, we did not conduct a meta-analysis. Two PCSs focused on encorafenib plus binimetinib therapy [67,68]; nevertheless, Menzies’s study was unsuitable for being included in a meta-analysis since the sample size encompassed only 13 individuals [68]. Therefore, the remaining eligible study was not meta-analyzed [67].

As a result, the meta-analysis was solely practicable for the vemurafenib monotherapy and dabrafenib plus trametinib regimens.

#### 3.2.1. Vemurafenib Monotherapy

The meta-analysis of RCTs on vemurafenib 960 mg showed a significant pooled prevalence of all-grade TRAEs of 18% (95% CI: 15–22%, *p* < 0.001; Q(26) = 471.96, *p* < 0.001, and I^2^ = 94.87%) [38,53,55,59,60]. The most common TRAEs are musculoskeletal and connective-tissue disorders (24%, 95% CI: 6–41%, *p* = 0.01; I^2^ = 97.59, *p* < 0.001), gastrointestinal disorders (22%, 95% CI: 13–33%, *p* < 0.001; I^2^ = 93.28, *p* < 0.001), and general disorders (defined here as constitutional toxicities, including fatigue, asthenia, and pyrexia; 20%, 95% CI: 17–22%, *p* < 0.001; I^2^ = 38.89, *p* = 0.15). Overall, arthralgia had the highest prevalence (44%, 95% CI: 29–59%, *p* < 0.001; I^2^ = 97.07, *p* < 0.001), followed by rash (39%, 95% CI: 22–56%, *p* < 0.001; I^2^ = 97.94, *p* < 0.001). Hyperkeratosis (29%, 95% CI: 24–33%, *p* < 0.001; I^2^ = 61.99, *p* < 0.001) and photosensitivity reaction (29%, 95% CI: 22–37%, *p* < 0.001; I^2^ = 86.28, *p* < 0.001) are relatively common toxicities. However, all these TRAEs are commonly of 1–2 grades. Squamous cell carcinoma is the most prevalent grade ≥ 3 TRAE (12%, 95% CI: 8–16%, *p* < 0.001; I^2^ = 81.80, *p* < 0.001), followed by rash (8%, 95% CI:4–13%, *p* < 0.001; I^2^ = 88.37, *p* < 0.001) and keratoacanthoma (6%, 95% CI: 2–9%, *p* < 0.001; I^2^ = 86.35, *p* < 0.001). Among the cardiac disorders, hypertension was found in 9% (95% CI: 7–12%, *p* < 0.001; I^2^ = 0.00, *p* < 0.001) of the sample (Table 2).

The limited number of PCSs on vemurafenib prevented us from performing a meta-analysis on TREAEs. Anforth et al. [62], Dika et al. [64], and Yamazaki et al. [73] were excluded due to the small sample sizes of ≤11 each. As a result, the only PCS eligible for a meta-analytic approach was not meta-analyzed [66].

#### 3.2.2. Dabrafenib Plus Trametinib

The meta-analysis of RCTs on dabrafenib plus trametinib indicated a significant pooled prevalence of all the grades of TRAEs of 14% (95% CI: 11–17%, *p* < 0.001; Q(36) = 630.23, *p* < 0.001, and I^2^ = 96.99%) [22,56,57,58,60,65]. The most common TRAEs are general disorders (defined here as constitutional toxicities, including fatigue, asthenia, and pyrexia; 25%, 95% CI: 14–37%, *p* < 0.001; I^2^ = 92.98, *p* < 0.001), followed by gastrointestinal disorders (20%, 95% CI: 12–27%, *p* < 0.001; I^2^ = 86.27, *p* < 0.001). Overall, fatigue is the most prevalent symptom (47%, 95% CI 38–56%, *p* < 0.001; I^2^ = 77.15, *p* < 0.001), followed by pyrexia (40%, 95% CI 26–54%, *p* < 0.001; I^2^ = 96.04, *p* < 0.001). All of these TRAEs, nevertheless, are typically of grades 1–2, registering very low incidences of grade ≥3 TRAEs (Table 3). Anemia registered a considerable prevalence (22%, 95% CI: 15–29%, *p* < 0.001; I^2^ = 51.40, *p* = 0.10). Among the cardiac disorders, hypertension and decreased left ventricular ejection registered prevalences of 11% (95% CI: 9–13%, *p* < 0.001; I^2^ = 5.57, *p* = 0.38) and 9% (95% CI: 7–11%, *p* < 0.001; I^2^ = 0.00, *p* = 0.60), respectively (Table 3).

The meta-analyses included two PCSs with a sample of 102 individuals [71,72]. The Anforth et al. and Dika et al. studies were excluded due to the small sample sizes [62,64], which were ≤10. A meta-analysis for all-grade TRAEs was feasible only for the following data: pyrexia (48%, 95% CI: 38–58%, *p* < 0.001; I^2^ = 0%, *p* < 0.001), aspartate aminotransferase (ASAT) (24%, 95% CI 15–32%, *p* < 0.001; I^2^ = 0%, *p* < 0.001), alkaline phosphatase (ALP) (24%, 95% CI 15–32%, *p* < 0.001; I^2^ = 0%, *p* < 0.001), and rash (19%, 95% CI 12–26%, *p* < 0.001; I^2^ = 0%, *p* < 0.001). Pyrexia was the only grade ≥3 computable outcome, accounting for a prevalence of 5% (95% CI 2–8%, *p* < 0.001; I^2^ = 0, *p* = 0.95).

### 3.3. Case Report Summary

Although some of the TRAEs were already reported in the clinical trial results, we present an overview of the published case reports to provide a comprehensive review of the available literature regarding toxicities related to BRAFis and MEKis. In total, we selected ten case reports published between 2012 and 2022 [75,76,77,78,79,80,81,82,83,84], involving individuals aged 20–70 with advanced BRAF mutant melanoma. The characteristics of the included cases are reported in Table 4.

### 3.4. Risk of Bias Assessment

Most of the studies showed a low risk of bias, except for open-label trials [38,55,57,60], which were found to have a moderate risk of bias, primarily due to the lack of blinding in the study design. A detailed domain-by-domain assessment is available in Appendix A.

The prospective cohort studies were evaluated using the ROBINS-I v2 tool. Within this group, the overall risk of bias ranged from serious to critical, primarily due to uncontrolled confounding factors, the absence of predefined protocols for outcome selection, and the potential for participant selection based on post-intervention characteristics. Only three studies were assessed as having a low risk of bias across all the domains [52,63,69], while one study was classified as having a moderate risk of bias [66]. A detailed evaluation of each domain is provided in Appendix A.

## 4. Discussion

This systematic review and meta-analysis provides a regimen-specific synthesis of the TRAEs associated with BRAF and MEK inhibitors in adults with BRAF-mutant CM, offering updated evidence focused on the most extensively studied targeted therapies. Our study addresses a key limitation of the prior meta-analyses [27,28,29], which often pooled heterogeneous melanoma subtypes and multiple treatment classes by restricting the analysis to cutaneous melanoma and separating monotherapy from combination regimens. This approach yields clinically actionable toxicity profiles that could inform both regimen selection and AE monitoring strategies. Overall, we observed that vemurafenib monotherapy was associated with a higher pooled prevalence of all-grade TRAEs (18%) than dabrafenib plus trametinib (14%), but the two regimens exhibited distinct toxicity phenotypes. These findings have implications for patient-tailored treatment decisions, particularly in the context of comorbidities and baseline functional status.

### 4.1. Vemurafenib Monotherapy

Vemurafenib is a selective inhibitor of the BRAF serine/threonine kinase, specifically targeting the V600E mutation found in melanoma cells. This drug was approved in August 2011 for the first-line treatment of unresectable or metastatic melanoma with the BRAF V600E mutation [38,85]. This drug targets and inhibits BRAF kinase activity, disrupting the MAPK signaling pathways and blocking the proliferation of malignant cells that carry this specific mutation [86]. Our results showed that the most common TRAEs included arthralgia (44%), rash (39%), diarrhea (33%), and vomiting (30%), which generally occurred at a low grade. Arthralgia is one of the most prevalent TRAEs affecting the small joints [87]. The hypothesis underlying arthralgia presentation is a paradoxical activation of the MAPK pathway in synovial tissue and lymphocytes by BRAF V600E inhibition [87]. Particular attention must be directed towards squamous cell carcinoma, the most prevalent grade ≥3 event, which exhibited comparable percentages to events of lower grades, followed by keratocantoma. Cancer skin toxicities, especially squamous cell carcinoma, are due to the paradoxical activation of the MAPK pathway in keratinocytes triggered by BRAFi therapy [88,89,90]. This toxicity mostly develops in the first three months of therapy, especially in sun-protected areas, with older adults at higher risk [88]. Precursor lesions, such as actinic keratoses, should be monitored during BRAFi therapy [89]. Accordingly, a substantial aggregate proportion of hyperkeratosis cases was found in our synthesis, which could represent a signal for timely interventions. Although the paradoxical activation of the MAPK pathway was also associated with rash occurrence, important differences were found in the mechanisms associated with rash, with some factors, such as female sex, representing a risk and others determining an incidence reduction, such as the combination with treatments inhibiting MEK, suggesting their potential role in TRAE treatment [87,91].

### 4.2. Dabrafenib Plus Trametinib

Dabrafenib (BRAFi) plus trametinib (MEKi) combination is a targeted therapy approved by the FDA in April 2018 for the treatment of stage III melanoma with a BRAF V600E or V600K mutation after surgery [92]. These drugs target the MAPK signaling pathway, which is overactive in melanoma cells due to BRAF mutations, and stop the proliferation of melanoma cells [93,94]. Dabrafenib selectively inhibits mutant BRAF, and trametinib selectively inhibits MEK1 and MEK2 proteins activated by RAF kinases [9]. While trametinib monotherapy is a treatment option, the combination with dabrafenib demonstrated higher response rates and longer median progression-free survival than dabrafenib monotherapy, with less cutaneous toxicity [10,95].

This combination showed higher benefits in terms of side effects and prognosis outcomes compared to vemurafenib monotherapy, supporting its use as the standard of care in this population [60,96]. Accordingly, our meta-analysis results indicated a lower TRAE burden of dabrafenib plus trametinib (14%) compared to vemurafenib monotherapy (18%). In this regard, multiple trials showed consistent health-related quality-of-life (HRQoL) advantages of combination BRAF and MEK inhibitors over BRAF inhibitor monotherapy [25]. The specific toxicological phenotype appeared to be different between the two treatment regimens as the most prevalent TRAEs with dabrafenib plus trametinib were fatigue (47%), pyrexia (40–48%), asthenia (39%), and chills (34%). The underlying mechanism of pyrexia during dabrafenib plus trametinib therapy is not yet fully understood. However, trametinib appears to influence the dabrafenib-driven pyrexia process by inducing interleukin-1 beta production in dendritic cells through BRAF inhibition, which leads to proinflammatory side effects [97,98]. The absence of pyrexia during trametinib monotherapy (also observed in our analysis) and the higher incidence of pyrexia in dabrafenib plus trametinib compared to dabrafenib monotherapy may support this hypothesis [99]. A similar underlying mechanism can be hypothesized for the high incidence of fatigue, which is generally driven by the activation of proinflammatory cytokines [100]. Furthermore, musculoskeletal and connective-tissue disorders, such as arthralgia, are less frequent with dabrafenib plus trametinib compared to vemurafenib. Additionally, immune-related disorders, including squamous cell carcinoma, skin papilloma, and hyperkeratosis, as well as skin conditions such as dry skin, erythema, alopecia, and photosensitivity, are less frequent with dabrafenib plus trametinib. However, MEKis were developed to block the paradoxical activation of the MAP kinase pathway, overcoming resistance and toxicity issues [101,102].

### 4.3. Insights from Case Reports

The results from case reports added knowledge to our comprehensive overview of TRAEs with BRAFis and MEKis by reporting some additional new TRAEs not documented in the meta-analyzed studies. One case of encephalitis was reported by Babacan et al. in 2021 [83], 13 days after initiation of vemurafenib plus cobimetinib, and flared up again 3 days after restarting BRAF inhibitor monotherapy with encorafenib. Although the patient previously received an anti-PD-1 antibody, with which cases of immune-mediated encephalitis were reported in the literature [103,104], the authors concluded that this effect was related to BRAF/MEK inhibitors for a clear temporal association between the symptom and therapy [83] and that the prior immunotherapy may have “primed” the immune system and predisposed the patient to this immune-related toxicity. This was the first reported case of encephalitis since, in a previous multicenter open-label randomized phase 3 trial testing encorafenib plus binimetinib versus vemurafenib or encorafenib, no similar cases were documented [55], nor in subsequent trials [66,67]. However, all the cohorts in these studies received prior immunotherapy. However, rare cases of grade ≥ 3 confusion were documented in some included clinical trials during both continuous and intermittent dabrafenib plus trametinib regimens [22,54]. In 2018, Loyson et al. reported the first case of hemorrhage in the liver on day 3 of dabrafenib plus trametinib [82]. A similar event was reported two years later by Algazi et al. [22] during a randomized phase 2 trial testing continuous versus intermittent doses of dabrafenib plus trametinib: one case of grade ≥ 3 gastric hemorrhages was registered during the continuous regimen. Two case reports documented granulomatous interstitial nephritis during dabrafenib plus trametinib therapy. In this regard, an undefined severe case of kidney injury was recorded in the Algazi phase-randomized trial during the intermittent regimen of dabrafenib plus trametinib [22]. The mechanism underlying all these events is the paradoxical activation of the MAPK pathway, which may activate T-cells and cause such immunomodulated toxicities [105].

### 4.4. Clinical Implications

The regimen-specific toxicity patterns identified in this meta-analysis should be actively integrated into clinical decision-making, with therapy selection guided not only by the expected efficacy but also by anticipated adverse event profiles and the patient’s comorbidity risk. Vemurafenib monotherapy may be less suitable for individuals with a history of multiple keratinocyte carcinomas, extensive actinic damage, or other pre-existing dermatologic conditions as our analysis showed a relatively high prevalence of cutaneous toxicities, including rash, hyperkeratosis, photosensitivity, and squamous cell carcinoma, likely related to paradoxical MAPK pathway activation in keratinocytes. In such patients, a baseline dermatologic assessment, timely treatment of precursor lesions, and scheduled skin surveillance may help to reduce the risk of severe or early-onset cutaneous malignancies during therapy.

Conversely, dabrafenib–trametinib, although associated with a lower prevalence of cutaneous neoplasms, was characterized in our analysis by a higher incidence of systemic inflammatory symptoms, such as pyrexia and fatigue, which may disrupt treatment continuity, as well as measurable rates of hypertension and a decreased left ventricular ejection fraction. These findings support close symptom monitoring and proactive pyrexia management, as well as periodic cardiac evaluation in patients with baseline cardiovascular disease or risk factors for cardiovascular diseases.

Incorporating these regimen-specific toxicity profiles into treatment discussions could enable more tailored monitoring strategies and facilitate early recognition and management of adverse events, supporting treatment adherence and optimizing patient outcomes. To translate these findings into consistent clinical practice, consensus documents and expert guidance are needed to provide more actionable evidence-based recommendations on monitoring, prevention, and management strategies tailored to each therapeutic regimen.

### 4.5. Limitations

Although this study provides important insights into the prevalence of TRAEs associated with target therapies in adults with BRAF-mutant CM, numerous limitations exist and need to be considered in the study’s findings. Most of these limitations are consistent with the study design and the applied methodological approach, which was conducted rigorously to minimize biases and provide reliable and attainable results.

Firstly, this study aggregates data derived from primary studies, which may be susceptible to computational errors. Furthermore, conducting a meta-analysis on TRAEs may be biased due to the assessment of whether or not an AE was related to the investigated treatment, leading to potential overestimation or underestimation of the proportions. However, these limits are inherent to the adopted data synthesis approach and align with the study’s aim. Secondly, we meta-analyzed studies with a relatively small sample size, which may result in inaccurate calculations and unreliable results. In this regard, we acknowledge that the study weight (determined by the number of cases and the sample size) significantly influences the estimation of the effect size. However, for proportional meta-analyses, there is no universally agreed-upon minimum sample size for individual studies; therefore, we attempt to minimize the effect size distortion due to small studies by including only studies with at least 25 individuals. Furthermore, there was significant variation in the sample sizes, which creates difficulty in detecting differences, especially when comparing RCT and PCS results and exploring different trends. Thirdly, highly significant statistical heterogeneities were observed across the estimations. The adoption of a random-effect model mitigated this limit. However, the I^2^ statistic should be interpreted with caution in meta-analyses of proportions as the high intrinsic clinical heterogeneity across prevalence studies renders the I^2^ statistic non-discriminative in proportional meta-analyses [45,106,107,108]. Nevertheless, we also observed several cases of significant statistical homogeneity, suggesting no real variations in the estimates of the effects across some of the combined studies. Fourthly, several rare TRAEs did not emerge from our synthesis (i.e., ocular toxicities, infections, and cardiac events), nor were some regimens analyzed, resulting in a limited overview of all the potential adverse events related to the treatments. However, our focus was to provide a quantitative synthesis of TRAEs when applicable rather than listing all the TRAEs already reported in the primary studies. Finally, some currently approved targeted therapy combinations, such as vemurafenib plus cobimetinib and encorafenib plus binimetinib, could not be comprehensively analyzed due to the limited number of eligible trials available at the time of our search. This further underscores the need for updated and well-powered clinical studies to fully characterize the safety profiles of these regimens.

## 5. Conclusions

Our study presents an interpretable picture of the most prevalent and investigated TRAEs associated with BRAF and MEK inhibitors in adults with CM based on trial design. These data are essential for quantifying treatment burden in light of advances in pharmacological and pharmaceutical science, which have enabled the development of new therapeutic classes, such as small-molecule kinase inhibitors. Because several trials had small sample sizes and reported TRAEs using heterogeneous definitions and measurement approaches, it was not possible to meta-analyze certain adverse events. This limitation highlights the inherent difficulty of quantifying treatment burden in this patient population.

The analyzed treatments have a distinct toxicity profile, with vemurafenib monotherapy being the most impactful treatment on symptom burden. A high prevalence of musculoskeletal and connective-tissue disorders (arthralgia at the top) was found during vemurafenib monotherapy. General disorders (mainly fatigue and pyrexia) were common during dabrafenib plus trametinib therapy. Cutaneous toxicities showed considerable prevalence during vemurafenib therapy, with rash as the most prevalent. Although TRAEs are commonly of low grades in all the treatment regimens, grade 3 or higher skin cancers (i.e., squamous cell carcinoma and keratocanthoma) occur quite frequently during vemurafenib treatment, with similar percentages compared to low-grade TRAEs. Gastrointestinal disorders reached similar percentages between vemurafenib monotherapy and the dabrafenib plus trametinib regimen. Although dabrafenib plus trametinib has a better overall safety profile than vemurafenib in terms of TRAE incidence, a broad spectrum of toxicities emerged from our analysis with the former. Despite the data on trametinib monotherapy being limited to a few small PCSs, treatment-related diarrhea seems to be highly prevalent. Cardiac toxicities, albeit infrequent, required specific monitoring in all the treatment regimens, with hypertension showing the highest prevalence across the treatments.

Although additional large-scale studies are needed to corroborate the findings, our study may contribute to the optimization of symptom management in clinical practice and may help to address healthcare costs. Standardized core outcome data sets are recommended in melanoma clinical trials to facilitate future research and provide a comprehensive overview of the TRAEs’ aggregated frequencies associated with BRAF and MEK inhibitors in adults with CM, especially for PCSs. Furthermore, regular dermatologic assessments throughout these treatments are highly recommended, along with a risk stratification assessment. A consensus on the monitoring and management of TRAEs, accompanied by clear recommendations, would be beneficial in optimizing TRAEs and ensuring a balance between risks and benefits.

## Figures and Tables

**Figure 1 cancers-17-03152-f001:**
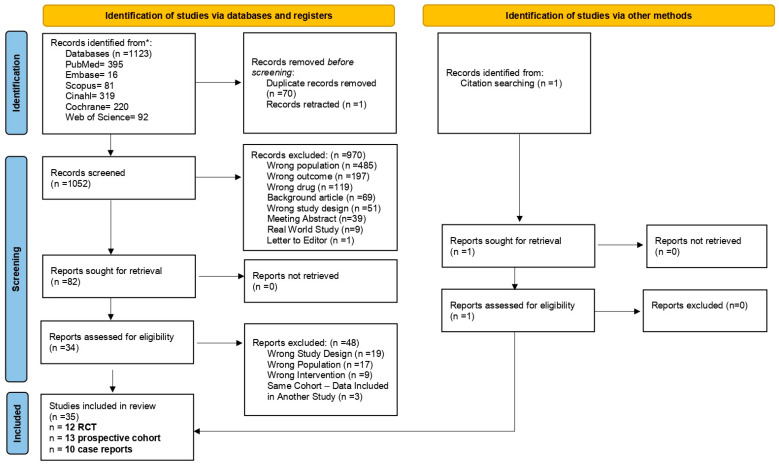
Flow diagram. * The databases were last searched on 28 February 2025.

**Table 1 cancers-17-03152-t001:** Characteristics of the studies.

First Author, Year	Country	Trial Phase and Name	Trial ID	Design	Sample	Treatment	Stage	AEs Assessment Scale and Coding
Chapman, 2011 [38]	Worldwide	Phase III	NCT01006980	RCT	675	Vemurafenib (n = 336)	IIIC–IV	CTCAE version 4
Flaherty, 2012 [65]	n.i.	Phase I/II	NCT01072175	PCS	247	Dabrafenib + Trametinib or Dabrafenib (n = 162)	IIIC–IV	CTCAE version 4
Kim, 2013 [74]	USA	Phase II	NCT01037127	PCS	97	Trametinib (n = 40)	IIIC	CTCAE version 3
Flaherty, 2014 [66]	USA	n.i.	n.i.	PCS	371	Vemurafenib (n = 371)	n.i.	CTCAE version 4
Johnson, 2014 [57]	Worldwide	Phase III	NCT01072175	RCT	383	Dabrafenib + Trametinib (n = 26) Dabrafenib (n = 45)	Stage III–IV	CTCAE version 4
Anforth, 2015 [62]	Australia	n.i.	n.i.	PCS	64	Vemurafenib (n = 11) Dabrafenib (n = 43) Dabrafenib + Trametinib (n = 10)	n.i.	n.i.
Yamazaki, 2015 [73]	Japan	Phase I/II	n.i.	PCS	11	Vemurafenib (n = 11)	Stage IIIC–IV	CTCAE version 4
Robert, 2015 [60]	Worldwide	Phase III	NCT01597908	RCT	704	Vemurafenib (n = 349)	Stage IIIC–IV	CTCAE version 4
Dika, 2016 [64]	Italy	n.i.	n.i.	PCS	24	Vemurafenib (n = 9) Dabrafenib (n = 5) Dabrafenib + Trametinib (n = 10)	n.i.	CTCAE version 4
Dréno, 2017 [53]	Worldwide	Phase III (coBRIM)	NCT01689519	RCT	493	Vemurafenib (n = 246)	n.i.	CTCAE version 4
Long, 2017 [58]	Worldwide	Phase III	NCT01682083	RCT	870	Dabrafenib + Trametinib (n = 438)	IIIA–IIIB–IIIC	n.i.
Schreuer, 2017 [71]	Belgium	Phase II	NCT02296996	PCS	25	Dabrafenib + Trametinib (n = 25)	IIIC–IV	CTCAE version 4
Maio, 2018 [59]	Worldwide	Phase III (BRIM8)	NCT01667419	RCT	498	Vemurafenib (n = 157) Vemurafenib (n = 93)	IIC–IIIA–IIIB	CTCAE version 4
Dummer, 2018 [55]	Worldwide	Phase III (COLUMBUS)	NCT01909453	RCT	577	Encorafenib + Binimetinib (n = 192) Encorafenib (n = 194) Vemurafenib (n = 191)	IIIB–IIIC–IV	CTCAE version 4.03
Robert, 2019 [52]	n.i.	Phase III (METRIC)	NCT01245062	RCT	322	Trametinib (n = 211)	IIIC–IV	n.i.
Algazi, 2020 [22]	USA	Phase II	NCT02196181	RCT	206	Dabrafenib + Trametinib (Intermittently) (n = 105) Dabrafenib +Trametinib (Continuously) (n = 101)	III–IV	CTCAE version 4
Ferrucci 2020 [61]	Worldwide	Phase II (KEYNOTE-022)	NCT02130466	RCT	120	Dabrafenib + Trametinib (n = 60)	III–IV	CTCAE version 4
Si, 2020 [72]	East Asia	Phase IIa	NCT02083354	PCS	77	Dabrafenib + Trametinib (n = 77)	IIIC–IV	CTCAE version 4.03
Ribas, 2020 [70]	USA	Phase Ib (BRIM7)	n.i.	PCS	129	Vemurafenib + Cobimetinib (n = 129)	IIIC–IV	CTCAE version 4
Nebhan, 2021 [69]	USA	Phase II	NCT02296112	PCS	9	Trametinib (n = 9)	III–IV	n.i.
Gonzalez-Cao, 2021 [56]	Spain	Phase II	NCT02583516	RCT	70	Vemurafenib (n = 35) Vemurafenib + Cobimetinib (n = 35)	IIIC–IV	CTCAE version 4.03
Awada, 2021 [63]	Belgium	Phase II	NCT04059224	PCS	16	Trametinib (n = 6)	IV-M1c	CTCAE version 4.03
Daymu, 2024 [54]	UK	Phase II (INTERIM)	n.i.	RCT	79	Dabrafenib + Trametinib (Intermittently) (n = 39) Dabrafenib + Trametinib (Continuously) (n = 40)	III–IV	CTCAE version 4.03
Màrquez-Rodas, 2024 [67]	Spain	Phase II (E-BRAIN/GEM1802)	NCT03898908	PCS	48	Encorafenib + Binimetinib (n = 48)	n.i.	CTCAE version 4
Menzies, 2024 [68]	n.i.	Phase II (POLARIS)	NCT03911869	PCS	13	Encorafenib + Binimetib (n = 13)	n.i.	n.i.

AEs (adverse events); RCT (randomized controlled trial); PCS (prospective cohort study); CTCAE (Common Terminology Criteria for Adverse Events); n.i.: no information.

**Table 2 cancers-17-03152-t002:** Prevalence of the most common TRAEs during vemurafenib (RCTs).

	Vemurafenib 960 mg					Vemurafenib 960 mg				
**n RCTs**	5					5				
**Total Sample**	1364					1364				
	**All grades**					**Grade ≥ 3**				
	**ES. CI (95%)**	** *p* **	**I^2^ (%)**	**n** **Studies**	**Sample Size**	**ES. CI (95%)**	** *p* **	**I^2^ (%)**	**n** **Studies**	**Sample Size**
**TRAEs**										
**Gastrointestinal disorders**										
Nausea [38,53,55,59,60]	0.28 (0.15–0.41)	<0.001	61.55 *	5	1364	0.01 (0.00–0.01)	<0.001	0.00	5	1364
Diarrhea [53,55,59,60]	0.33 (0.28–0.38)	<0.001	65.98 *	4	1028	0.01 (0.00–0.02)	0.03	49.92	4	1028
Vomiting [53,55,59,60]	0.14 (0.12–0.17)	<0.001	0.00	4	1028	0.01 (0.00–0.02)	0.01	0.00	4	1028
Decreased appetite [53,55,59]	0.17 (0.13–0.22)	<0.001	61.29	3	679	0.01 (0.00–0.01)	0.12	0.00 *	3	679
**Total**	0.22 (0.13–0.33)	<0.001	93.28 *							
**General disorders**										
Fatigue [38,53,55,59]	0.27 (0.16–0.38)	<0.001	94.47 *	4	1015	0.02 (0.01–0.03)	<0.001	0.00	4	1015
Asthenia [55,59]	0.17 (0.14–0.21)	<0.001	0.00 *	2	433	0.02 (0.01–0.04)	0.01	0.00 *	2	433
Pyrexia [53,55,59,60]	0.22 (0.18–0.26)	<0.001	57.08	4	1028	N.a.	/	/	/	/
Headache [55,59]	0.19 (0.16–0.23)	<0.001	0.00 *	2	433	N.a.	/	/	/	/
**Total**	0.20 (0.17–0.22)	<0.001	38.89							
**Metabolic disorders**										
Increased ALT [53,55,59]	0.14 (0.07–0.21)	<0.001	86.49 *	3	679	0.04 (0.01–0.07)	0.01	78.35 *	3	679
Increased ASAT [53,59]	0.12 (0.09–0.15)	<0.001	0.00 *	2	493	0.03 (0.01–0.04)	<0.001	0.00 *	2	493
Increased GGT [53,55,59]	0.11 (0.02–0.19)	0.02	93.58 *	3	679	0.05 (0.02–0.09)	0.01	81.53 *	3	679
**Total**	0.12 (0.10–0.15)	<0.001	0.01							
**Skin tissue and immune-related disorders**										
Alopecia [38,53,55,59,60]	0.30 (0.15–0.46)	<0.001	97.91 *	5	1364	N.a.	/	/	/	/
Rash [38,53,55,59,60]	0.39 (0.22–0.56)	<0.001	97.94 *	5	1364	0.08 (0.05–0.12)	<0.001	85.02 *	5	1364
Maculopapular rash [55,59]	0.10 (0.08–0.13)	<0.001	0.00 *	2	433	0.03 (0.01–0.04)	<0.001	0.00 *	2	433
Erythema [55,59]	0.16 (0.12–0.19)	<0.001	0.00 *	2	433	N.a.	/	/	/	/
Pruritus [38,55,59]	0.15 (0.04–0.27)	<0.001	95.74 *	3	769	0.01 (0.00–0.02)	0.01	0.00 *	2 [37,59]	583
Dry skin [55,59]	0.21 (0.17–0.25)	<0.001	0.00 *	2	433	N.a.	/	/	/	/
Hyperkeratosis [53,55,59,60]	0.29 (0.24–0.33)	<0.001	61.99 *	4	1028	0.01 (0.00–0.02)	0.02	0.00 *	4	1028
Skin papilloma [55,59,60]	0.19 (0.14–0.23)	<0.001	61.51	3	782	N.a.	/	/	/	/
Keratoacanthoma [38,53,55,59]	0.09 (0.07–0.11)	<0.001	0.00	4	1178	0.06 (0.02–0.09)	<0.001	86.35 *	4	1015
Squamous cell carcinoma [53,59,60]	0.12 (0.08–0.17)	<0.001	82.15 *	4	1178	0.12 (0.08–0.16)	<0.001	81.80 *	4	1178
**Total**	0.18 (0.13–0.23)	<0.001	93.09 *							
**Ocular toxicity**										
Photosensitivity reaction [53,55,59,60]	0.29 (0.22–0.37)	<0.001	86.28 *	4	1028	0.01 (0.00–0.02)	0.08	35.00	4	1028
**Musculoskeletal and connective-tissue disorders**										
Pain in the extremities [55,59]	0.16 (0.13–0.20)	<0.001	0.00 *	2	433	N.a.	/	/	/	/
Arthralgia [37,53,55,59,60]	0.44 (0.29–0.59)	<0.001	97.07 *	5	1364	0.05 (0.03–0.06)	<0.001	12.23	5	1364
Myalgia [55,59]	0.15 (0.12–0.18)	<0.001	0.00 *	2	433	0.01 (0.00–0.02)	0.06	0.00 *	2	433
**Total**	0.24 (0.06–0.41)	0.01	97.59 *							
**Cardiac disorders**										
Increased CPK [53,55]	0.02 (0.01–0.04)	<0.001	0.00 *	2	432	0.01 (0.00–0.01)	0.08	0.00 *	2	432
Hypertension [55,59]	0.09 (0.07–0.12)	<0.001	0.00 *	2	433	0.03 (0.01–0.04)	<0.001	0.00 *	2	433
**Total**	0.05 (0.01–0.12)	0.12	95.48 *							
**TOTAL**	**0.18 (0.15–0.22)**	**<0.001**	**94.87 ***							

n (number); RCTs (randomized controlled trials); TRAEs (treatment-related adverse events); ES (effect size); CI 95% (95% confidence interval); GGT (Gamma-Glutamyl Transferase); ASAT (aspartate aminotransferase; ALT (alanine aminotransferase); CPK (creatine phosphokinase); N.a. (not applicable): insufficient number of studies to perform a meta-analysis or no cases or negligible incidences that lead to a negative lower bound. * Significance at *p* ≤ 0.05. Note: General disorders = constitutional toxicities, including fatigue, asthenia, pyrexia, headache, chills, influenza-like illness, and peripheral edema, as classified under the CTCAE category ‘General disorders and administration site conditions’.

**Table 3 cancers-17-03152-t003:** Prevalence of the most common TRAEs during dabrafenib plus trametinib (RCTs).

	Dabrafenib 150 mg Twice Daily Plus Trametinib 1–2 mg Daily					Dabrafenib 150 mg Twice Daily Plus Trametinib 1–2 mg Daily				
**n RCTs**	6					7				
**Total sample**	1140					1238				
	**All grades**					**Grade ≥ 3**				
	**ES. CI (95%)**	** *p* **	**I^2^ (%)**	**n** **studies**	**Sample size**	**ES. CI (95%)**	** *p* **	**I^2^ (%)**	**n** **studies**	**Sample size**
**TRAEs**										
**Gastrointestinal disorders**										
Nausea [56,65]	0.25 (0.05–0.43)	0.01	86.59 *	2	125	0.01 (0.00–0.01)	0.02	0.00	6	1026
Diarrhea [22,56,57,58,60,65]	0.27 (0.22–0.33)	<0.001	70.44 *	6	1160	0.01 (0.01–0.02)	<0.001	0.00	5 [22,54,56,58,60]	964
Vomiting [56,57,58,60,65]	0.26 (0.20–0.32)	<0.001	67.81 *	5	955	0.01 (0.00–0.02)	<0.001	0.00	5 [54,56,58,60,65]	998
Decreased appetite [56,57,58,65]	0.12 (0.09–0.15)	<0.001	0.00	3	570	N.a.	/	/	/	/
Constipation [56,57,58,65]	0.12 (0.05–0.19)	<0.001	78.14 *	4	570	N.a.	/	/	/	/
**Total**	0.20 (0.12–0.27)	<0.001	86.27 *							
**General disorders**										
Fatigue [22,57,58,65]	0.47 (0.38–0.56)	<0.001	77.15 *	4	740	0.04 (0.03–0.05)	<0.001	0.00	5 [22,54,57,58,65]	818
Asthenia [56,58]	0.39 (0.09–0.51)	0.01	89.72 *	2	505	N.a.	/	/	/	/
Pyrexia [22,56,57,58,60,65]	0.40 (0.26–0.54)	<0.001	96.04 *	6	1160	0.04 (0.02–0.06)	<0.001	48.72	6	1168
Headache [56,57,58,65]	0.20 (0.03–0.37)	0.02	96.19 *	4	605	N.a.	/	/	/	/
Chills [22,57,58,60,65]	0.324(0.28–0.41)	<0.001	77.73 *	5	1090	0.01 (0.01–0.02)	<0.001	0.00	5 [22,54,58,60,65]	1123
Dizziness [56,57]	0.09 (0.03–0.15)	<0.001	0.00 *	2	115	N.a.	/	/	/	/
Influenza-like illness [22,58]	0.10 (0.02–0.18)	0.01	89.67 *	2	640	N.a.	/	/	/	/
Peripheral edema [22,56,57,58,65]	0.09 (0.04–0.14)	0.01	88.54 *	5	810	N.a.	/	/	/	/
**Total**	0.25 (0.14–0.37)	<0.001	92.98 *							
**Metabolic disorders**										
Increased ALT [22,56,58]	0.18 (0.12–0.24)	<0.001	69.02 *	3	675	0.03 (0.02–0.05)	<0.001	0.00	2 [22,58]	640
Increased ASAT [22,58]	0.15 (0.12–0.19)	<0.001	30.15	2	640	0.02 (0.01–0.04)	0.01	52.06	2 [22,58]	640
Increased ALP [22,58,65]	0.15 (0.06–0.23)	<0.001	79.01 *	3	295	N.a.	/	/	/	/
Increased lipase [22,56]	0.09 (0.02–0.15)	0.01	70.62 *	2	240	0.03 (0.01–0.06)	<0.001	0.00	2 [22,56]	239
Increased amylase [22,56]	0.05 (0.02–0.08)	<0.001	0.00	2	240	N.a.	/	/	/	/
**Total**	0.12 (0.07–0.17)	<0.001	80.88 *							
**Skin tissue and immune-related disorders**										
Alopecia [56,60,65]	0.06 (0.04–0.08)	<0.001	0.00	3	440	N.a.	/	/	/	/
Rash [56,57,58,60,65]	0.23 (0.20–0.26)	<0.001	0.00	5	955	0.01 (0.00–0.02)	0.02	0 *	2 [56,60]	385
Maculopapular rash [22,56]	0.13 (0.06–0.20)	<0.001	68.79 *	2	275	N.a.	/	/	/	/
Erythema [56,58]	0.08 (0.02–0.14)	<0.001	70.25 *	2	505	N.a.	/	/	/	/
Dry skin [22,56,58]	0.07 (0.03–0.12)	<0.001	73.81 *	3	710	N.a.	/	/	/	/
Dermatitis acneiform [56,60,65]	0.09 (0.04–0.14)	<0.001	77.73 *	3	855	N.a.	/	/	/	/
Hyperkeratosis [56,60,65]	0.04 (0.03–0.06)	<0.001	0.00	3	440	N.a.	/	/	/	/
Skin papilloma [56,60,65]	0.02 (0.01–0.03)	<0.001	0.00	3	440	N.a.	/	/	/	/
Squamous cell carcinoma [56,57,60,65]	0.04 (0.00–0.07)	0.03	45.53	4	485	N.a.	/	/	/	/
**Total**	0.08 (0.04–0.13)	<0.001	96.34 *							
**Ocular toxicity**										
Photosensitivity reaction [56,60]	0.09 (0.00–0.18)	0.05	77.43 *	2	420	N.a.	/	/	/	/
**Musculoskeletal and connective-tissue disorders**										
Back pain [22,57]	0.05 (0.02–0.09)	<0.001	0.00 *	2	149	N.a.	/	/	/	/
Arthralgia [22,56,58,60,65]	0.20 (0.15–0.26)	<0.001	76.06 *	5	1115	0.01 (0.00–0.02)	<0.001	0.00	4 [22,56,58,60]	1025
Myalgia [22,56,58,65]	0.12 (0.08–0.17)	<0.001	63.02 *	4	730	N.a.	/	/	/	/
Muscle weakness [22,56]	0.05 (0.02–0.07)	<0.001	0.00	2	240	0.02 (0.00–0.04)	0.03	0.00	2 [22,56]	240
**Total**	0.10 (0.03–0.17)	<0.001	92.36 *							
**Cardiac disorders**										
Hypertension [22,56,57,58,65]	0.11 (0.09–0.13)	<0.001	5.57	5	810	0.04 (0.02–0.06)	<0.001	10.72	4 [56,57,58,65]	605
QT prolongation [22,56]	0.03 (0.01–0.05)	0.01	0.00	2	240	N.a.	/	/	/	/
Decreased LVEF [22,57,60,65]	0.09 (0.07–0.11)	<0.001	0.00	4	655	0.04 (0.02–0.05)	<0.001	0.00	3 [22,57,60]	600
**Total**	0.08 (0.03–0.12)	<0.001	93.99 *							
**Blood and lymphatic system** **disorders**										
Anaemia [22,56,57]	0.22 (0.15–0.29)	<0.001	51.40	3	285	0.02 (0.00–0.04)	0.01	0.00	2 [22,57]	250
Neutropenia [22,54]	N.a.	/	/	/	/	0.02 (0.00–0.04)	0.03	22.74	2	283
Lymphopenia [22,56]	0.10 (0.02–0.17)	0.01	75.44 *	2	240	N.a.	/	/	/	/
**Total**	0.16 (0.04–0.28)	0.01	80.97 *							
**Infections**										
Sepsis [22,54]	N.a.	/	/	/	/	0.02 (0.00–0.04)	0.05	0.00	2	182
**TOTAL**	**0.14 (0.11–0.17)**	**<0.001**	**96.99 ***							

Legend: n (number); RCTs (randomized controlled trials); TRAEs (treatment-related adverse events); ES (effect size); CI 95% (95% confidence interval); ASAT (aspartate aminotransferase); ALT (alanine aminotransferase); ALP (alkaline phosphatase); CPK (creatine phosphokinase); LVEF (left ventricular ejection fraction); N.a. (not applicable): insufficient number of studies to perform a meta-analysis or no cases or negligible incidences that lead to a negative lower bound. * Significance at *p* ≤ 0.05. These meta-analyses contain both continuous and intermittent regimen data. Note: Starting doses for trametinib varied across trials (1 mg/day or 2 mg/day, with 2 mg/day being the standard in the U.S.). Where available, data on dose-limiting toxicities (DLTs) and treatment discontinuations due to toxicity are reported. In most trials, toxicities reflect the cumulative patient experience during therapy rather than events occurring only at full doses prior to DLT. General disorders = constitutional toxicities, including fatigue, asthenia, pyrexia, headache, chills, influenza-like illness, and peripheral edema, as classified under the CTCAE category ‘General disorders and administration site conditions’.

**Table 4 cancers-17-03152-t004:** Case report summary.

First Author, Year	Disease	Melanoma Stage	Metastatic	Treatment	Patient Age	Patient Sex	AEs
Zimmer, 2012 [75]	BRAF mutant melanoma	IV	Yes	Vemurafenib (960 mg, orally twice daily) Dabrafenib (150 mg, orally twice daily)	44 59	Female Female	Panniculitis with arthralgia Panniculitis with arthralgia
Chen, 2014 [76]	n.i.	n.i.	Yes	Vemurafenib + cobimetinib	20	Female	Involution of eruptive melanocytic nevi
Orouji, 2014 [77]	BRAF mutant melanoma	n.i.	Yes	Vemurafenib (960 mg, orally twice daily)	64	Female	Leukopenia and neutropenia
Park, 2014 [78]	BRAF mutant melanoma	IV	Yes	Dabrafenib (75 mg, orally twice daily) Vemurafenib (960 mg, orally twice daily)	80 70	Female Male	Perifollicular granulomatous inflammation Erythematous and violaceous papules
Jansen, 2015 [79]	BRAF mutant melanoma	IIIa	Yes	Dabrafenib (150 mg, orally twice daily) + trametinib (2 mg once daily)	61	Male	Granulomatous nephritis and dermatitis
Carrera, 2015 [80]	BRAF mutant melanoma	IIIc	n.i.	Dabrafenib	30	Female	Multiple BRAF Wild-Type melanomas
Keating, 2016 [81]	BRAF mutant melanoma	n.i.	Yes	Dabrafenib (150 mg, orally twice daily) + trametinib (2 mg once daily)	64	Male	Robust and curly hair growth
Loyson, 2018 [82]	BRAF mutant melanoma	IIIc	Yes	Dabrafenib (150 mg, orally twice daily) + trametinib (2 mg once daily)	45	Female	Hemorrhage in the liver and bone metastasis
Babacan, 2021 [83]	BRAF mutant melanoma	IIIa	Yes	Vemurafenib + trametinib and later encorafenib	60	Male	Encephalitis
De Rick, 2022 [84]	n.i.	n.i.	Yes	Dabrafenib + trametinib	50	Male	Interstitial nephritis

## Data Availability

Data sharing is not applicable to this article. All data are available in the text and as Appendix A.

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
