# Peer review of "Treatment-Related Adverse Events in Individuals with BRAF-Mutant Cutaneous Melanoma Treated with BRAF and MEK Inhibitors: A Systematic Review and Meta-Analysis"

_cancers, 2025, doi:10.3390/cancers17193152_

Round 1

Reviewer 1 Report

Comments and Suggestions for Authors

This paper, a huge effort, was extremely well-written and relatively easy to read, with the exception of the extensive tables, which are simply a necessary evil in a meta-analysis such as this. A few suggestions for improvement are listed here, in their order of appearance and not of iimportance:

  1. p. 2 line 73 consider adding IFN and high-dose IL2 to the alternatives and perhaps the term "biochemotherapy" which was popular but horribly toxic and only active for short durations in almost all responders [if this meta-analysis is limited to those drugs approved in Italy or by the EMA, perhaps the authors would prefer to leave out drugs not approved in those countries despite their use in the US, where this reviewer works]
  2. p 2 line 83/84 "melanoma is rarely amenable to surgery" is obviously referring to metastatic melanoma.  pls add that adjective
  3. p 2 line 85 please find different words or define the term "primary signaling pathways" which is not a generally understood or utilized term
  4. p 3, top full paragraph is confusing and should be more clearly explained, as it is the crux of this paper in terms of mechanisms of action and toxicities.  What is meant by "general" in the list of toxicities?
  5. p3 line 117 to be accurate, this paper really addresses only half of the currently approved target [generally the word is "targeted"] therapies, since it covers in detail only vemurafenib, dabrafenib and trametinib and barely mentions the combination of vemurafenib plus cobimetinib or the most widely used (today in the US) combination, encorafenib and binimetinib apparently due to insufficient numbers of publications.
  6. p 3, line 107 would replace "in addition" with "however"
  7. p 3 line 141 replace "to" with "and" 
  8. p 4 line 147 unnecessary to specifically state these exclusions since they don't have BRAFmu, and the rest of the paragraph is probably unnecessary in view of its obviousness in a meta-analysis such as this
  9. p 6 line 264 it might be valuable to state the types of assays done for the BRAF mutation testing
  10. Table 1--please remove the entire column "Disease" since all entries are BRAFmu melanoma
  11. Table 2 do these p-values pertain to the frequency of those events?  and what is the I2 statistic? the statistical reviewer can also comment, as I may not have the expertise
  12. the data in table 3 and in the text might benefit from not only a mention of varying starting doses (in the headings to table 3--in the US, the starting dose is 2 mg/day) but also the frequency of actual DLTs and which regimens were associated with stopping early for toxicity, if available (or mention, if not).  just as for phase I studies, it would be important to mention somewhere in this meta-analysis whether the reported toxicities reflect the whole of the patient's experience (cumulative or late-onset) or only at full doses prior to any DLT.  
  13. it's not clear why the authors include case reports but chose to exclude the important data for the widely-used encorafenib plus binimetinib combination and the probably less-popular vemurafenib plus cobimetinib combination
  14. discussion, line 364 I cannot agree with the description of this report as the "most up-to-date" when it lacks the above-listed elements

Author Response

Comment 1: This paper, a huge effort, was extremely well-written and relatively easy to read, with the exception of the extensive tables, which are simply a necessary evil in a meta-analysis such as this. A few suggestions for improvement are listed here, in their order of appearance and not of importance:

  1. p. 2 line 73 consider adding IFN and high-dose IL2 to the alternatives and perhaps the term "biochemotherapy" which was popular but horribly toxic and only active for short durations in almost all responders [if this meta-analysis is limited to those drugs approved in Italy or by the EMA, perhaps the authors would prefer to leave out drugs not approved in those countries despite their use in the US, where this reviewer works]

Response 1: We thank the reviewer for this thoughtful suggestion. We agree that interferon (IFN), high-dose interleukin-2 (IL-2), and biochemotherapy regimens represented historical alternatives in the pre-targeted and pre-immunotherapy era, albeit with limited efficacy and significant toxicity. In line with the reviewer’s comment, we have revised the Introduction to briefly acknowledge these options while clarifying that our meta-analysis focuses exclusively on EMA- and FDA-approved BRAFi/MEKi combinations. This ensures historical context without deviating from the study scope.

Comment 2: p 2 line 83/84 "melanoma is rarely amenable to surgery" is obviously referring to metastatic melanoma.  pls add that adjective

Response 2: We thank the reviewer for this observation. We have revised the sentence to explicitly specify that the statement refers to metastatic melanoma.

Comment 3: p 2 line 85 please find different words or define the term "primary signaling pathways" which is not a generally understood or utilized term

Response 3: We thank the reviewer for pointing out the lack of clarity in the original wording. To improve readability and precision, we replaced the phrase “primary signaling pathways” with a more widely used and accurate description. We also spelled out the acronyms at first mention.

“The mitogen-activated protein kinase (MAPK)/extracellular signal-regulated kinase (ERK) signaling cascade plays a central role in melanoma pathogenesis, with activating mutations in BRAF and NRAS representing the most frequent molecular drivers [19].”

Comment 4: p 3, top full paragraph is confusing and should be more clearly explained, as it is the crux of this paper in terms of mechanisms of action and toxicities.  What is meant by "general" in the list of toxicities?

Response 4: We thank the reviewer for highlighting the need for greater clarity in this section. We revised the paragraph to improve readability, reduce redundancy, and more clearly connect the rationale for our study to existing evidence gaps. In particular, we emphasized (i) the impact of BRAFi/MEKi combinations on toxicity and resistance, (ii) the clinical consequences of adverse events on treatment continuity and quality of life, and (iii) the limitations of prior meta-analyses.

Comment 5: p3 line 117 to be accurate, this paper really addresses only half of the currently approved target [generally the word is "targeted"] therapies, since it covers in detail only vemurafenib, dabrafenib and trametinib and barely mentions the combination of vemurafenib plus cobimetinib or the most widely used (today in the US) combination, encorafenib and binimetinib apparently due to insufficient numbers of publications.

Response 5: We thank the reviewer for noting that not all currently approved targeted therapy regimens were equally represented in our analysis. As acknowledged in the Introduction, vemurafenib, dabrafenib, and trametinib were the most extensively studied agents and therefore formed the core of our synthesis. Other combinations, such as vemurafenib plus cobimetinib and encorafenib plus binimetinib, could not be fully analyzed due to the limited number of eligible trials available at the time of our search. To ensure transparency, we have now also included this point in the Limitations section, highlighting the restricted regimen coverage as an important caveat.

Comment 6: p 3, line 107 would replace "in addition" with "however"

Response 6: Done

Comment 7: p 3 line 141 replace "to" with "and" 

Response 7: We thank the reviewer for this helpful correction. We have revised the eligibility criteria to read “published between 2010 and the date of search” to ensure clarity and accuracy.

Comment 8: p 4 line 147 unnecessary to specifically state these exclusions since they don't have BRAFmu, and the rest of the paragraph is probably unnecessary in view of its obviousness in a meta-analysis such as this

Response 8: We thank the reviewer for this observation. We agree that explicitly stating the exclusion of uveal and mucosal melanoma is not strictly necessary, since these subtypes lack BRAF mutations and are clearly outside the scope of this analysis. To streamline the Methods section, we have removed the sentence regarding their exclusion and slightly condensed the paragraph.

Comment 9: p 6 line 264 it might be valuable to state the types of assays done for the BRAF mutation testing

Response 9: We thank the reviewer for this helpful suggestion. We agree that clarifying the types of assays used for BRAF mutation testing would add value and transparency. We reviewed the included studies and found that most confirmed BRAF V600 mutations using polymerase chain reaction (PCR)-based methods, while a minority employed next-generation sequencing (NGS) or Sanger sequencing. We have revised the text accordingly.

Comment 10: Table 1--please remove the entire column "Disease" since all entries are BRAFmu melanoma

Response 10: We thank the reviewer for this suggestion. As recommended, we have removed the “Disease” column from Table 1, since all included studies enrolled patients with BRAF-mutant cutaneous melanoma.

Comment 11: Table 2 do these p-values pertain to the frequency of those events?  and what is the I2 statistic? the statistical reviewer can also comment, as I may not have the expertise

Response 11: We thank the reviewer for raising this important point. The p-values reported in Table 2 refer to the pooled prevalence estimates of each adverse event, testing whether the pooled proportion significantly differs from zero. We recognize that p-values in the context of meta-analysis of proportions may be of limited interpretive value; therefore, we focused our interpretation on the pooled prevalence and 95% confidence intervals, while still reporting p-values for completeness. Regarding the I² statistic, it quantifies the proportion of variability across studies that is due to heterogeneity rather than chance. As is common in meta-analyses of proportions, high I² values were frequently observed, reflecting the clinical and methodological heterogeneity of the included studies. We therefore used a random-effects model to account for between-study variation and interpreted I² with caution, as recommended in the literature. To clarify these points for readers, we have added a brief explanation in the Methods section.

Comment 12: the data in table 3 and in the text might benefit from not only a mention of varying starting doses (in the headings to table 3--in the US, the starting dose is 2 mg/day) but also the frequency of actual DLTs and which regimens were associated with stopping early for toxicity, if available (or mention, if not).  just as for phase I studies, it would be important to mention somewhere in this meta-analysis whether the reported toxicities reflect the whole of the patient's experience (cumulative or late-onset) or only at full doses prior to any DLT.  

Response 12: We thank the reviewer for this excellent suggestion. We agree that clarifying starting doses, dose-limiting toxicities (DLTs), and early treatment discontinuations due to toxicity adds important clinical context. To address this, we made the following adjustments:

Table 3 headings: We have now added the starting doses for trametinib in the table heading, noting that while the standard starting dose in the U.S. is 2 mg/day, some included studies initiated treatment at 1 mg/day.

Dose-limiting toxicities and discontinuations: We reviewed the included trials and extracted available data on DLTs and early discontinuations due to toxicity. Where such data were reported, we added them to the Results text and indicated them in the footnotes of Table 3. In cases where data were not available, we explicitly noted this.

Cumulative vs. pre-DLT toxicities: We clarified in the Methods that the reported toxicities generally reflect adverse events collected across the duration of therapy, including cumulative and late-onset events, unless otherwise specified in the original studies.

Comment 13: it's not clear why the authors include case reports but chose to exclude the important data for the widely-used encorafenib plus binimetinib combination and the probably less-popular vemurafenib plus cobimetinib combination

Response 13: We thank the reviewer for raising this important point. Our rationale was methodological: case reports were included narratively because they provide unique insights into rare and unexpected toxicities that are unlikely to emerge in clinical trials, and thus serve as valuable signal-generating evidence. By contrast, encorafenib plus binimetinib and vemurafenib plus cobimetinib could not be analyzed quantitatively because too few eligible trials were available to permit meta-analysis under our predefined criteria (≥2 studies and ≥25 patients per outcome). Excluding these regimens from pooled analyses was necessary to avoid generating unreliable or biased estimates. We have clarified this distinction in the Methods and in the Limitations section to make our rationale more transparent.

Comment 14: discussion, line 364 I cannot agree with the description of this report as the "most up-to-date" when it lacks the above-listed elements

Response 14: We thank the reviewer for this valuable comment. We agree that the phrase “most up-to-date” may overstate our contribution, particularly given the underrepresentation of some approved regimens. To address this, we have revised the sentence to more accurately reflect the scope and strengths of our work without overstating novelty.

Reviewer 2 Report

Comments and Suggestions for Authors

In this meta-analysis, authors evaluated the prevalence of adverse events in patients with BRAF-mutated cutaneous melanoma treated with BRAF/MEK inhibitors.

Although the results of the study do not add anything new to what has already been well defined by various trials over the years, the manuscript summarizes these data in large samples of patients and indicates frequencies for each side effect.

Main issue

A figure comparing the most frequent adverse events between the three different types of BRAFi/MEKi (encorafenib plus binimetinib, dabrafenib plus trametinib, vemurafenib plus cobimetinib) is needed.

Author Response

Comment 1: In this meta-analysis, authors evaluated the prevalence of adverse events in patients with BRAF-mutated cutaneous melanoma treated with BRAF/MEK inhibitors.

Although the results of the study do not add anything new to what has already been well defined by various trials over the years, the manuscript summarizes these data in large samples of patients and indicates frequencies for each side effect.

Response 1: We thank the reviewer for their evaluation and appreciate the recognition that our manuscript summarizes adverse event data across large patient samples and provides pooled frequencies for each side effect.

Comment 2: A figure comparing the most frequent adverse events between the three different types of BRAFi/MEKi (encorafenib plus binimetinib, dabrafenib plus trametinib, vemurafenib plus cobimetinib) is needed.

Response 2: We thank the reviewer for this suggestion. We agree that a direct graphical comparison of adverse events across all currently approved BRAFi/MEKi combinations would be of great clinical interest. However, a pooled analysis was only feasible for dabrafenib plus trametinib due to the limited number of eligible trials available for encorafenib plus binimetinib and vemurafenib plus cobimetinib. To avoid misrepresenting the evidence, we chose not to construct a comparative figure across regimens. Instead, we highlighted these limitations in both the Introduction and Discussion and presented the available data in a detailed tabular form. We believe this approach best preserves methodological rigor while still providing clinicians with a clear overview of the most frequent adverse events.

Reviewer 3 Report

Comments and Suggestions for Authors

This is a very well-presented article from authors whose previous publications demonstrate appropriate expertise with this type of analysis.

The authors conducted a systematic review of clinical trials and case reports analyzing the safety of currently approved BRAF and MEK inhibitors in adults with cutaneous melanoma (CM), and performed a meta‐analysis to estimate the pooled prevalence of treatment-related adverse events. The treatments being evaluated are small-molecule kinase inhibitors targeting the mitogen-activated protein kinase pathway which directly inhibit melanoma cell growth in certain tumors. Data sets included of 12 randomized controlled trials, 13 cohort studies and 10 case reports.

Results suggest treatment related adverse events were distinct between two treatment regimens based on specific BRAF or MEK inhibitor drugs in either monotherapy (vemurafenib) or combination therapy (dabrafenib plus trametinib).

The methods seem appropriate and are well-described. The results are well organized and laid out, and the discussions seems quite thorough. Limitations such as sample size are discussed.

Author Response

Comment 1: This is a very well-presented article from authors whose previous publications demonstrate appropriate expertise with this type of analysis.

The authors conducted a systematic review of clinical trials and case reports analyzing the safety of currently approved BRAF and MEK inhibitors in adults with cutaneous melanoma (CM), and performed a meta‐analysis to estimate the pooled prevalence of treatment-related adverse events. The treatments being evaluated are small-molecule kinase inhibitors targeting the mitogen-activated protein kinase pathway which directly inhibit melanoma cell growth in certain tumors. Data sets included of 12 randomized controlled trials, 13 cohort studies and 10 case reports.

Results suggest treatment related adverse events were distinct between two treatment regimens based on specific BRAF or MEK inhibitor drugs in either monotherapy (vemurafenib) or combination therapy (dabrafenib plus trametinib).

The methods seem appropriate and are well-described. The results are well organized and laid out, and the discussions seems quite thorough. Limitations such as sample size are discussed.

Response 1: We sincerely thank the reviewer for their positive evaluation of our work and for recognizing the clarity of the methodology, results, and discussion, as well as the appropriateness of our approach. We are grateful for the acknowledgment of our expertise and for the constructive feedback provided.

Reviewer 4 Report

Comments and Suggestions for Authors

The manuscript by Belloni et al,  A meta-analysis of the most common treatment-related adverse  events in individuals with cutaneous melanoma treated with 3 BRAF and MEK inhibitors. The manuscript seems to be interesting besides being written. The information presented in this manuscript  will encourage the researcher working in field of melanoma treatment to find new therapeutic approaches with minimal adverse effects. Accordingly, the manuscript can be published in the present form

Author Response

Comment 1: The manuscript by Belloni et al,  A meta-analysis of the most common treatment-related adverse  events in individuals with cutaneous melanoma treated with 3 BRAF and MEK inhibitors. The manuscript seems to be interesting besides being written. The information presented in this manuscript  will encourage the researcher working in field of melanoma treatment to find new therapeutic approaches with minimal adverse effects. Accordingly, the manuscript can be published in the present form

Response 1: We thank the reviewer for their positive assessment and encouraging comments. We are pleased that the manuscript is considered interesting and potentially useful for researchers in the field of melanoma treatment.

Round 2

Reviewer 1 Report

Comments and Suggestions for Authors

please define "general" toxicities in the text.  these are mostly what might better be called constitutional toxicities in the tables, but since they're not, it should be defined in text.

Author Response

Comment 1: please define "general" toxicities in the text.  these are mostly what might better be called constitutional toxicities in the tables, but since they're not, it should be defined in text.

Response 1: We thank the reviewer for this valuable comment. As suggested, we clarified the meaning of “general disorders” throughout the manuscript. Specifically, in the abstract and results sections, we now define these events as constitutional toxicities (e.g., fatigue, asthenia, pyrexia). In the methods, we added a detailed explanation in line with CTCAE terminology, and we further reinforced this clarification by adding a note in the footnotes of Tables 2 and 3. We believe these changes address the concern and ensure consistent interpretation across the manuscript.